# Synergistic Effect of Biphasic Calcium Phosphate and Platelet-Rich Fibrin Attenuate Markers for Inflammation and Osteoclast Differentiation by Suppressing *NF-κB*/*MAPK* Signaling Pathway in Chronic Periodontitis

**DOI:** 10.3390/molecules26216578

**Published:** 2021-10-30

**Authors:** Anil Kumar, Jaideep Mahendra, Little Mahendra, Hesham H. Abdulkarim, Mohammed Sayed, Maryam H. Mugri, Zeeshan Heera Ahmad, Ashok Kumar Bhati, Hadeel Hussain Faqehi, Waleed Omar Algregri, Saranya Varadarajan, Thodur Madapusi Balaji, Hosam Ali Baeshen, Shankargouda Patil

**Affiliations:** 1Department of Periodontology, Meenakshi Ammal Dental College and Hospital, Meenakshi Academy of Higher Education and Research, Chennai 600095, India; dranilkumar7979@yahoo.com; 2Maktoum Bin Hamdan Dental University College, Dubai 213620, United Arab Emirates; littlemahendra24@gmail.com; 3Advanced Periodontal and Dental Implant Care, Missouri School of Dentistry and Oral Health, A. T. Still University, St. Louis, MO 63104, USA; Heshamabdulkarim@atsu.edu; 4Department of Prosthetic Dental Sciences, College of Dentistry, Jazan University, Jazan 45412, Saudi Arabia; drsayed203@gmail.com; 5Department of Maxillofacial Surgery and Diagnostic Sciences, College of Dentistry, Jazan University, Jazan 45412, Saudi Arabia; dr.mugri@gmail.com; 6Dental College Hospital, King Saud University Medical City, Riyadh 12372, Saudi Arabia; aheera@ksu.edu.sa; 7Division of Periodontics Department of Preventive Dental Sciences, College of Dentistry, Jazan University, Jazan 45412, Saudi Arabia; gums_ashh@yahoo.com; 8Dental School, Jazan University, Jazan 45412, Saudi Arabia; hadeel937@iCloud.com; 9King Fahad Armed Forces Hospital, Jeddah 23311, Saudi Arabia; Algregri394@gmail.com; 10Department of Oral Pathology and Microbiology, Sri Venkateswara Dental College and Hospital, Chennai 600130, India; vsaranya87@gmail.com; 11Department of Periodontology, Tagore Dental College and Hospital, Chennai 600127, India; tmbala81@gmail.com; 12Department of Orthodontics, College of Dentistry, King Abdulaziz University, Jeddah 22254, Saudi Arabia; drbaeshen@me.com; 13Department of Maxillofacial Surgery and Diagnostic Sciences, Division of Oral Pathology, College of Dentistry, Jazan University, Jazan 45412, Saudi Arabia

**Keywords:** osteoclastogenesis, MAPK, NF-kB, periodontitis, platelet-rich fibrin, biphasic calcium phosphate

## Abstract

Background: Periodontitis is characterized by excessive osteoclastic activity, which is closely associated with inflammation. It is well established that MAPK/NF-kB axis is a key signaling pathway engaged in osteoclast differentiation. It is stated that that biphasic calcium phosphate (BCP) and platelet-rich fibrin (PRF) have significant antiostoeclastogenic effects in chronic periodontitis. Objective: We aimed to elucidate the synergetic effect of PRF/BCP involvement of the nuclear factor kappa–light–chain–enhancer of activated B cells (NF-kB) and the mitogen-activated protein kinase (*MAPK*) signaling pathway in osteoclast differentiation in chronic periodontitis. Methods: We induced osteoclast differentiation in vitro using peripheral blood mononuclear cells (PBMCs) derived from patients with chronic periodontitis. We assessed osteoclast generation by tartrate-resistant acid phosphatase (TRAP) activity, proinflammatory cytokines were investigated by ELISA and NF-κB, and *IKB* by immunoblot, respectively. MAPK proteins and osteoclast transcription factors were studied by Western blot analysis and osteoclast transcriptional genes were assessed by RT-PCR. Results: The results showed that the potent inhibitory effect of PRF/BCP on osteoclastogenesis was evidenced by decreased TRAP activity and the expression of transcription factors, NFATc1, c-Fos, and the osteoclast marker genes, TRAP, MMP-9, and cathepsin-K were found to be reduced. Further, the protective effect of PRF/BCP on inflammation-mediated osteoclastogenesis in chronic periodontitis was shown by decreased levels of proinflammatory cytokines, NF-kB, IKB, and MAPK proteins. Conclusions: PRF/BCP may promote a synergetic combination that could be used as a strong inhibitor of inflammation-induced osteoclastogenesis in chronic periodontitis.

## 1. Introduction

Periodontitis is an infectious chronic inflammatory disease which is caused by periodontal pathogenic bacteria affecting the periodontal tissues leading to tooth loss [1]. Despite eliciting an immune response, periodontal tissues also primarily provoke the host inflammatory response to *Porphyromonas gingivalis,* involving the recruitment of inflammatory cells, production of cytokines, osteoclast activation, and bone resorption [2].

The inflamed periodontium consists of various immune and nonimmune cells that interact with bacterial constituents such as LPS to generate interleukins, such as IL-1β, IL-6, and TNF-α which act as an autocrine factor to cause chronic periodontal inflammation. In periodontal inflammation, elevated levels of inflammatory mediators are found in tissues leading to alveolar bone resorption [3].

Inflammatory gene expression is induced by intracellular signal transduction mechanisms. Inflammatory signaling pathways are controlled by mitogen-activated protein kinases (*MAPK*) from the cell surface to the nucleus [4]. Three well-characterized subfamilies of *MAPK* pathways include: (1) The c-Jun N-terminal kinase (*JNK*) pathway; (2) the extracellular signal-related kinase (*ERK1/2*) pathway; and (3) the p38 *MAPK* pathway.

*MAPK* activates various transcription factors such as activator protein-1 (*AP-1*) and NF-kB, substrate proteins for phosphorylation, including the “downstream” cytoskeletal elements, nuclear receptors, cell death receptors, and serine/threonine kinases [5]

Cell differentiation and movement, gene expression, embryogenesis, mitosis, metabolism, programmed death, and many other cellular activities are regulated by *MAPK* [6]. The p38 *MAPK* pathway is vital to signal stress, inflammatory, and infectious stimuli; and it is also involved in the control of essential processes including cell proliferation, differentiation, and migration [7]. The c-Jun N-terminal kinases (*JNK*s) belong to a family of stress-activated protein kinases that are associated with the transactivation of c-Jun by phosphorylating the N-terminal serine residues [8]. Growth factors and inflammatory cytokines also activate *JNK*s [9]. *JNK*s and ERKs regulate cell proliferation and cell death.

*MAPK* are implicated in the regulation of osteoclast activation along with the expression of inflammatory mediators. Osteoclast differentiation is mainly regulated by the interplay of RANKL/RANK [10]. Rapid phosphorylation and activation of *MAPK* are mainly regulated by RANKL, and this activates transcription factors such as a nuclear factor of activated T cell 1 (*NFATc1*), *c-Fos*, *OPG* and tumor necrosis factor receptor-associated factor-6 (TRAF 6), thereby regulating the gene expression for osteoclast differentiation [11].

The application of regenerative materials has shown a promising result on osteoclast differentiation [12]. However, the role of PEF/BCP in inhibiting the *MAPK* signaling pathway has not been evaluated thus far.

Platelet concentrate, most widely known as platelet-rich fibrin (PRF) is derived from autologous blood which is rich in growth factors. PRF is extensively used in the management of periodontal intrabony defects, furcation defects, maxillofacial surgical procedures, and regenerative procedures [12]. Zhang et al. demonstrated the favorable effects of PRF membranes in alveolar ridge preservation [13]. PRF facilitates angiogenesis, cell proliferation, and differentiation, which in turn leads to new bone and tissue regeneration [14]. In addition, a previous study reported an impact of leukocyte and platelet-rich fibrin (L-PRF) exudate on bone regeneration when incorporated with the poly(Lactide-co-glycolide) (PLGA) [15]. Recent studies also stated that the formation of osteoclasts can be inhibited with PRF membranes from hematopoietic progenitors in bone marrow cultures [16,17].

In the era of tissue engineering and regeneration, biphasic calcium phosphate has emerged and contributed immensely. According to Silva et al., BCP has shown a great impact on the migration of macrophages and secretion [18]. Osteointegration and bone formation in periodontal defects were noted when biphasic calcium phosphate was used synthetically as fabricated bone graft [19]. One of the advantages of calcium phosphate is its osteoinductive and osteoconductive property which aids in the osteogenic differentiation of mesenchymal stem cells [20,21]. Balaguer et al. demonstrated that plasma clotted around BCP microparticles showed an osteogenic property that could be used for the treatment of bony defects [22].

We recently demonstrated in vitro that in chronic periodontitis patients, the ostoeoclastic effect was inhibited through the promotion of the proteolytic cascade of apoptosis when treated with PRF/BCP [23]. In this study, we hypothesized that inhibiting the *MAPK*/*NF-kB* signaling strategy would provide a novel antiosteoclastogenic target in periodontitis progression. Earlier studies described the role of several natural *MAPK* pathway inhibitors such as myricetin, luteolin, morin, fisetin, and panduratin A from medicinal plants on osteoclastic differentiation and periodontal inflammation [24,25]. No past studies have reported the molecular effect of PRF and BCP on influencing the *MAPK*/NF-Kbaxis, IL-6, IL-1β, TNF-α, transcription factors, and osteoclastic marker genes

Hence, in the present study, we aimed to elucidate the synergistic effect of PRF/BCP on the markers for inflammation and osteoclast differentiation by suppressing the *MAPK*/*NF-kB* axis in chronic periodontitis in vitro.

## 2. Materials and Methods

### 2.1. Materials and Reagents

Bone BCP (SigmaGraft, CA, USA) is comprised of 40% beta-tricalcium phosphate (β-TCP) and 60% hydroxyapatite (HA). The reagents included were 3-(4,5-dimethylthiazol-2-yl)-2,5-diphenyltetrazolium bromide (MTT), Dulbecco’s Modified Eagle’s Medium (DMEM), TRI reagent, trypan blue. All other chemicals of research-grade were obtained from Sigma-Aldrich, Inc (St. Louis, MO, USA). The heat-inactivated fetal bovine serum (FBS) was procured from GIBCO Grand Island, New York, USA. RANKL was obtained from Sigma-Aldrich (St. Louis, MO, USA) and M-CSF from BioVision, (Milpitas, CA, USA). The bicinchoninic acid (BCA) protein assay kit was purchased from Thermo Fisher Scientific (Rockford, IL, USA) and the primary antibodies (1:1000 dilutions) *NF-κB*, *NFATc1*, *c-Fos*, TRAF 6, *ERK, p-ERK*, *JNK*, p-*JNK*, *P38,* and *p-P38* were supplied by Santa Cruz, CA, USA. Secondary antibody, β-actin was purchased from Santa Cruz (USA).

### 2.2. Study Design

The patients were recruited from the Department of Periodontology during the period from July 2015 until February 2016. Fifteen generalized chronic periodontitis individuals, both male and female, within the age range of 35–45 years were selected according to the criteria mentioned below. The human subject’s ethics board of MAHER, Chennai, India approved the study (“Institutional Review Board”, Protocol No: MU-128-IEC-2015). The research was conducted at par with the 1975 Declaration of Helsinki 1975, as revised in 2013. The written consent was obtained from the subjects who were willing to participate in the study.

### 2.3. Inclusion and Exclusion Criteria 

According to the criteria, both male and female patients aged 35–45 years with more than 5 mm of clinical attachment loss in 30% or more sites with at least 20 teeth remaining in the oral cavity were included as generalized chronic periodontitis (AAP 1999 classification) patients and became part of our investigation [26]. These patients were nonsmokers and had no previous history of other systemic diseases. Moreover, they were also not treated for chronic periodontitis for the past 6 months. Patients under antibiotic and anti-inflammatory therapy were excluded from the study. From the selected chronic periodontitis (CP) patients, the PBMCs were isolated and divided into four groups based on the treatment with PRF/BCP. The first group (CP) remained untreated, the second group (CP+BCP) was treated with BCP alone, the third group (CP+PRF) was treated with PRF alone, and the fourth group (CP+PRF/BCP) was treated with PRF/BCP in combination.

### 2.4. Preparation of PRF and BCP

Five ml venous blood from chronic periodontitis subjects was collected and transferred to 10 mL sterile glass centrifuge tubes with centrifugation at 3000 rpm for 12 min (R-4C, REMI, Maharashtra, India). In the middle of the tube, a fibrin clot was formed between the red corpuscles and acellular plasma. The serum was squeezed out from the clot using sterile gauze, and a resistant autologous platelet-rich fibrin membrane was obtained. The membrane was then cut into fragments. PRF membranes were then minced to a size of 1 × 1 cm for the further analyses. The BCP grafting particles, composed of 60% hydroxyapatite and 40% β-tricalcium phosphate were purchased commercially (SigmaGraft, CA, USA).

The minced PRF threads were added to BCP and the PRF/BCP mixture was used for treating the cells. The PRF/BCP mixture was prepared as 1 mL/well using the DMEM medium and added to the wells. BCP (60 µg/mL) was dissolved in stored acellular plasma and the culture medium was used to adjust the concentration of the drug.

### 2.5. Monocytes Isolation from Whole Blood

Five ml of peripheral blood was collected from the antecubital vein for analysis from chronic periodontitis patients. Mononuclear cells were obtained from diluted peripheral blood (1:2 in phosphate-buffered saline (PBS)), which was layered over Ficoll–Paque (Sigma-Aldrich, Inc., USA), and it was further centrifuged at 1700 rpm for 30 min at room temperature. The solution was then washed and resuspended in Dulbecco’s Modified Eagle Medium (DMEM) (Sigma-Aldrich Inc., USA) containing 10% fetal bovine serum (FBS) (GIBCO Grand Island, New York, NY, USA). Consequently, these cells were calculated in a hemocytometer using trypan blue.

### 2.6. Osteoclasts Generation and Differentiation

PBMCs were plated in 24-well plates at a density of 1 × 10^5^ cells per well in 2 mL of DMEM medium, containing 10% FBS (FBS; HyClone), penicillin (100 µg/mL), and streptomycin (100 µg/mL) and allowed to adhere overnight. The next day, the medium was semidepleted and replaced with a fresh osteoclastogenic differentiation medium containing 25 ng/mL M-CSF (BioVision, Milpitas, CA, USA), 40 ng/mL RANKL (Sigma-Aldrich Inc., USA), 1 μM dexamethasone, and 2 mm L-glutamine. The cells were allowed to be re-fed twice a week by withdrawing half of the medium and replacing it with fresh. The cells were harvested, separated, and then divided into four different groups where BCP, PRF, and PRF/BCP mixtures were added in six parallel wells, the CP group was the control and they were cultured for 21 days and stained for tartate resistant acid phosphatase (*TRAP*) activity. All the experiments were performed in triplicate for error elimination and authenticity of the results.

### 2.7. TRAP Activity Assay

Ten μL of cell lysate was added and incubated at 37 °C for 30 min to the 50 μL *TRAP* reaction buffer (2.5 naphthol ASBI phosphate HCl in 100 mM sodium acetate and 50 mM disodium tartrate pH 6.1). To stop the enzymatic reaction finally, 150 μL of 0.1 M NaOH was added. A multifunction microplate reader was used to measure the fluorescence with the wavelength of 405/520 nm. In all four groups, calibrator solutions with different *TRAP* concentrations (0.6 to 12 U/L) (Bone *TRAP*, Medac, Wedel, Germany), were used to correlate the fluorescence intensity with *TRAP* activity.

### 2.8. Measurement of Proinflammatory Cytokines by ELISA

ELISA kits {R&D Systems, Inc (Minneapolis, MN, USA)} were used to analyze the levels of IL-1β, IL-6, and TNF-α, and they were determined in the osteoclast culture medium based on the manufacturer’s directions.

### 2.9. Western Blot Analysis

The cells from four different experimental groups were plated in 24-well plates at a density of 5 × 10^4^ cells per well in 1 mL of DMEM containing 10% FBS overnight. After 24 h, nonadherent cells were removed by gentle washing. The biomaterials, namely BCP, PRF, and the combination of PRF and BCP, were added to the cells. After 24 h of treatment, cell lysis was conducted by adding a cold RIPA buffer (150 mM NaCl, 50 mM Tris HCL, 0.1% SDS, 1% Triton X-100, 1 mM PMSF, 2 mM NaF, Na_3_VO_4_, β-glycerophosphate, 2 mM EDTA, and fresh protease inhibitor cocktail), and the cell lysate was centrifuged at 14,000 rpm at 4 °C for 20 min. The BCA method was used to analyze the protein content of the supernatant. Protein was denatured in a sample buffer, separated on 12% SDS-PAGE, and was then transferred to a polyvinylidene difluoride membrane. The blots were blocked for 2 h at room temperature with Tris-buffered saline (TBS, 50 mM Tris-HCl, pH 7.5, 150 mM NaCl) containing 5% nonfat milk. The blots were then washed three times with TBST (50 mM Tris-HCl, pH 7.5, 150 mM NaCl, and 0.02% Tween 20) and incubated with a specific primary antibody (1:1000 dilutions) of *NF-κB, IKB*, *NFATc1*, *c-Fos*, *ERK, p-ERK*, *JNK*, p-*JNK*, *P38,* and *p-P38* (Santa Cruz, CA, USA) at 4 °C and kept overnight. The blots were then incubated for 1 h at room temperature with secondary antibody (1:5000 dilutions), and detected by an enhanced chemiluminescence (ECL) (Thermo Fisher Scientific) detection reagent. β-actin was used as an internal control to ensure that equal amounts of sample proteins were applied for electrophoresis. Densitometric analysis was performed using Image Lab software (Bio-Rad Laboratories, Hercules, CA, USA).

### 2.10. RNA Isolation and Reverse Transcription-Polymerase Chain Reaction

According to the manufacturer’s instructions, in all four groups, total RNA was isolated from cells by a TRIzol reagent (Invitrogen, Carlsbad, CA, USA). cDNA was synthesized from 1 µg of RNA using an iScript cDNA Synthesis Kit. (Bio-Rad Laboratories, Hercules, CA, USA). PCR amplifications were performed as follows: 30 cycles for *TRAP* (95 °C for 60 s, 55 °C for 30 s, and 60 °C for 60 s), 30 cycles for *CAT-K* (94 °C for 1 min, 60 °C for 1 min, and 72 °C for 1 min) (ENST00000678337.1), 40 cycles for *MMP-9* (94 °C for 60 s, 60 °C for 60 s, and 68 °C for 120 s), and 30 cycles for β-actin (94 °C for 35 s, 64 °C for 45 s, and 72 °C for 1 min) (ENSG00000075624). β-actin was used as an endogenous control. Ethidium bromide was used to stain the PCR products and then analyzed in a 1% agarose gel. A 100 bp ladder was used to confirm the size of the amplification products. Relative mRNA expression levels were obtained by normalizing to the β-actin expression. All the experiments were repeated three times. Primers for the study are represented in Table 1.

### 2.11. Statistical Analysis

Statistical package social sciences version (SPSS) 23.0 was used to perform the statistical analysis. All the data in this study was presented as mean ± standard deviation (SD). Differences among groups were assessed using one way ANOVA and Dunnett’s multiple comparison tests for intergroup comparison was conducted using Graphpad Prism 6.0 software package for windows. *p* < 0.05 was considered to be statistically significant. * *p* < 0.05, ** *p* < 0.01, *** *p* < 0.001.

## 3. Results

### 3.1. PRF/BCP Decreases TRAP Activity

The activity of *TRAP* was significantly suppressed by PRF/BCP as compared with other groups (PRF, BCP and CP) (Figure 1). Hence, PRF/BCP might efficiently prevent osteoclastogenesis by inhibiting *TRAP*.

### 3.2. PRF/BCP Attenuates Proinflammatory Cytokines

IL-1β, IL-6, and TNF-α were significantly decreased when treated with PRF/BCP as compared with PRF and BCP alone (Figure 2). In addition, as compared with other cytokines, the level of IL-6 expression was markedly decreased.

### 3.3. PRF/BCP Regulates Periodontal Inflammation and Osteoclastogenesis by Inhibiting MAPK Signaling and NF-κB Pathways

*MAPK* signaling and *NF-kB* pathways were examined at the protein level on osteoclasts. In the chronic periodontitis group (CP), *MAPK* signaling proteins and their phosphorylated forms were highly expressed (Figure 3a,b). PRF/BCP markedly inhibited the expression of *ERK, p-ERK*, *JNK*, p-*JNK*, *p-38, and p-p38* and also diminished the expression of NF-kB and *IKB* (Figure 3c,d). These results indicate that PRF/BCP has antiosteoclastogenic and anti-inflammatory effects which are regulated via the inhibition of *MAPK* signaling proteins.

### 3.4. PRF/BCP Inhibits Osteoclastogenic Transcription Factors and Osteoclast Marker Genes

Osteoclastic transcription factors, *c-Fos*, *NFATc1*, and *TRAF 6* and bone resorptive enzymes, *cathepsin K, TRAP,* and *MMP-9* were significantly decreased when treated with PRF/BCP (Figure 4a,b and Figure 5a,b). The decrease was greater in the PRF/BCP combined group than the PRF and BCP group alone. These results indicated the antiosteoclastic effect of PRF/BCP was established through inhibition of osteoclast-related marker genes.

## 4. Discussion

In the existing study, PRF along with BCP, was identified as an important negative modulator of markers for inflammation and osteoclast differentiation in chronic periodontitis [23]. This has led to the foundation for further research to explore the antiosteoclastic effect of PRF/BCP via *NF-kB* and *MAPK* signaling pathways.

The monocytes were derived from the peripheral blood and differentiated to form multinucleated osteoclasts [27]. According to Kumar et al., *TRAP* staining showed that there was a relative decrease in the number of *TRAP* positive multinucleated osteoclasts in PRF+BCP combination (*p* < 0.001) as compared with other groups [23]. In the present study, *TRAP* activity on the osteoclastic cells also found a significant suppression of osteoclastic cells when treated with PRF/BCP, which is further confirmation of our previous study (Figure 1).

In general, it is hypothesized that IL-1β, IL-6, and TNF-α play a key role in activating osteoclasts that terminate in net bone resorption [28]. In the present study, IL-1β, IL-6, and TNF-α were increased in the CP group, suggesting an inflammatory response of the inflammatory cells during the progression of the disease (Figure 2). There were consistent results in different studies demonstrating the increased levels of proinflammatory cytokines in inflamed periodontal tissue [29,30]. It is observed that inflammatory responses are inhibited by the antagonist of IL-1 and TNF-α, thereby decreasing the amount of bone destruction in experimentally induced periodontitis [31].

*NF-kB* is one of the most common inflammatory cytokines and plays a key role in RANKL-induced osteoclast formation [32]. In the current study, NF-kB level was significantly increased in the CP group (Figure 3c,d), which can be explained as NF-kB binds to the promoter regions of proinflammatory cytokine genes and activates their transcription, thereby regulating the inflammatory process which is mainly mediated by *MAPK* pathway. The results also showed that PRF/BCP suppressed the osteoclast differentiation through the inhibition of *IKB* phosphorylation (Figure 3c,d) in the *NF-kB* signaling pathway.

*MAPK*, extracellular signal-regulated kinase (*ERK1/2*), c-Jun N-terminal kinase (*JNK*), and p38 mainly activate *NF-kB*, resulting in the expression of proinflammatory cytokines. However, PRF/BCP significantly decreased the level of proinflammatory cytokines (Figure 2) and *NF-kB* (Figure 3c,d), thus protecting against periodontal inflammation.

Moreover, considering that periodontal inflammation accelerates osteoclastogenesis, the function of *MAPK* signaling molecules in regulating osteoclast differentiation in periodontitis was further studied. RANKL is overexpressed by the inflammatory responses, which result in RANK receptor binding on an osteoclast, thereby causing osteoclastogenesis [33,34]. We observed an increased level of p38, *ERK1/2*, and *JNK* in the CP group where PRF/BCP was not introduced (Figure 3a,b). On the other hand, in PRF/BCP group, osteoclastogenesis was inhibited by antagonizing the *MAPK* signaling pathway (Figure 3a,b). Our study was in agreement with a study conducted by Lee et al. who concluded that osteoclastogenesis is mainly influenced by the *MAPK* pathway [35]. Another study also concluded that pharmacological inhibitors and gene silencing can also inhibit the *MAPK* pathway [36]. Previous studies also reported the importance of p38 in the regulation of IL-6 in periodontal ligament fibroblasts and osteoblasts [37,38]. This implies that the activation of the *MAPK* pathway might exist in periodontitis-induced osteoclastogenesis. Kim et al. showed the effect of panduratin A on inhibiting the *MAPK* signaling pathway in periodontitis [24]. The study also demonstrated the key aspects of p38, *JNK*, and ERK signals, which play a pivotal role in the differentiation of osteoclasts and intracellular signaling transduction.

In the present study, PRF/BCP affectively reduced the expression of transcription factors such as *NFATc1* and *c-Fos* in osteoclast differentiation in the PRF/BCP group as compared with PRF and BCP alone (Figure 4a,b). Our findings were in accordance with Choi et al., who described the role of fisetin as a potential inhibitor of RANKL-induced osteoclast differentiation [39]. Another study by Boyce et al. demonstrated that a decrease in signaling by p38/*c-Fos*/*NFATc1* could subsequently lead to a reduction in the expression of genes required for bone resorption, which is similar to our study [40]. It is assumed that during RANKL and RANK interaction, the transcription factors combine to the target gene within the nucleus of the osteoclastic cells. Further, phosphorylated *MAPK* signaling proteins activate *NFATc1* and *c-Fos*. Thus, the phosphorylation of *NFATc1* through the *MAPK*/*NF-kB* axis is critical for the master osteoclast transcription factor, *NFATc1*, to translocate into the nucleus of the osteoclast progenitor cells and activate the osteoclastogenic mechanism. In our study, the synergistic effect of PRF/BCP showed to attenuate the action of the transcription factors, thereby influencing the *MAPK* signaling proteins and reducing the osteoclastic activity.

We also observed that in response to the transcriptional signal, the expression of mRNA in osteoclastic specific genes such as *TRAP*, cathepsin K, and MMP-9 were significantly inhibited by PRF/BCP (Figure 5a,b). Our findings were in agreement with Franco et al., who described the inhibitory effect of doxycycline on the functional expression of osteoclast markers genes, such as *TRAP* activity, *MMP9* enzyme activity, and cathepsin K on RANKL-induced osteoclastogenesis [41]. Hence, keeping the aforementioned concepts in mind, it can be concluded that PRF/BCP can reduce the manifestation and regulation of osteoclastic transcription factors by regulating the *MAPK* signaling pathway proteins, whereby blocking the activation of *TRAP*, cathepsin K, and MMP-9. One of the most important factors that exhibits the bioactive feature of BCP is the degradation and release of calcium ions which also regulate the formation and resorption of osteoclasts [21,42]. On the other hand, phosphates and the osteoblastic lineage via the IGF-1 and *ERK1/2* pathways regulate the growth and differentiation of osteoblasts, thereby increasing the expression of BMPs [43,44]. Hence, BCP has a dual role to play as a regenerative material.

Phosphate mainly plays an important role in the regulation of the RANK ligand: OPG ratio to inhibit osteoclast differentiation and bone resorption [45]. The expression of osteoblastic differentiation markers such as *ALP*, *BMPs*, *OPN*, *OCN*, *BSP*, *ON*, and *RunX2* is also affected by calcium and phosphate ions [46,47]. However, the mechanism by which PRF/BCP induces osteogenesis in favor of bone formation remains to be established. The findings of this study can also pave the way to understanding new treatment modalities to manage periodontal diseases. Further research should be undertaken to realize the importance of signaling mechanisms in inflammatory mediators, their interactions with the *MAPK*/*NF-kB* signaling pathway, and also to obtain more predictable clinical results.

### Limitations of the Current Study

This study has certain limitations. First, the study identified osteoclasts using *TRAP* activity; however, pit assay can be used to determine the bone resorption activity of osteoclasts. Secondly, there was a lack of healthy groups which could provide more insight concerning the *TRAP* activity, *MAPK* pathway, transcription factors, and osteoclast marker genes when treated with PRF and BCP in comparison with the other groups. Furthermore, the investigations were restricted to only one time point. Various study designs at varying time points would enable better inference of the study. In the future, a large number of studies with more samples may ensure better significant findings with the best precise outcome in estimating the clinical effects of PRF/BCP combination in regenerative periodontics.

## 5. Conclusions

Overall, this study demonstrated the inhibitory effect of PRF and BCP on the activation of *NF-kB* and *MAPK* signaling pathways, as well as on the expression of inflammatory and osteoclastogenic genes. The study results provide vital evidence substantiating the potential role of a PRF and BCP combination as a regenerative medicine in the therapeutic approach to periodontitis. As the present study showed the involvement of multiple signaling pathways and the crosstalk between PRF/BCP and signaling modulators, future studies on the biologic roles of other signaling pathways/modulators activated by PRF/BCP might provide further understanding of biomaterials as novel therapeutic agents in the treatment of chronic periodontitis.

## Figures and Tables

**Figure 1 molecules-26-06578-f001:**
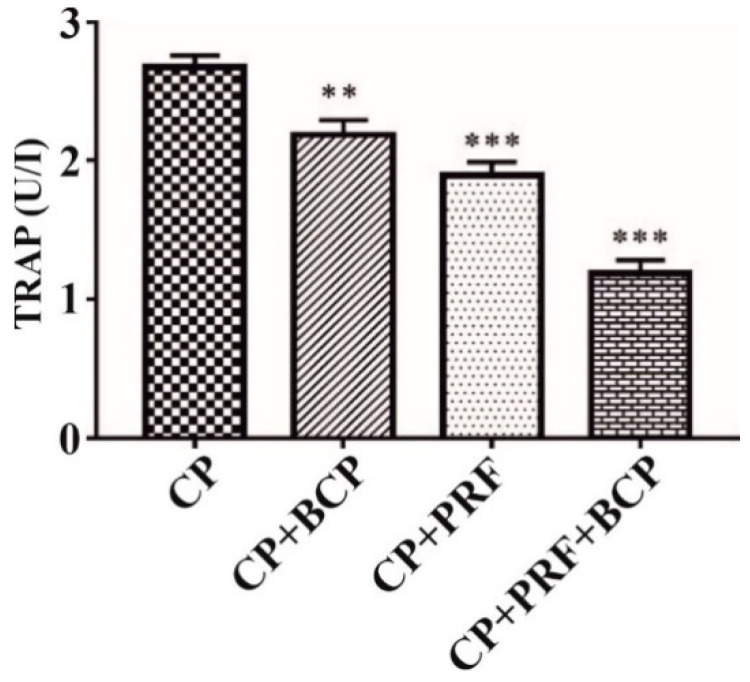
*TRAP* activity images. All data are presented by mean ± SD. Results are expressed as the mean ± SD. The comparisons were made among CP and BCP, CP and PRF, and CP and PRF/BCP. (** *p*< 0.01; *** *p* < 0.001).

**Figure 2 molecules-26-06578-f002:**
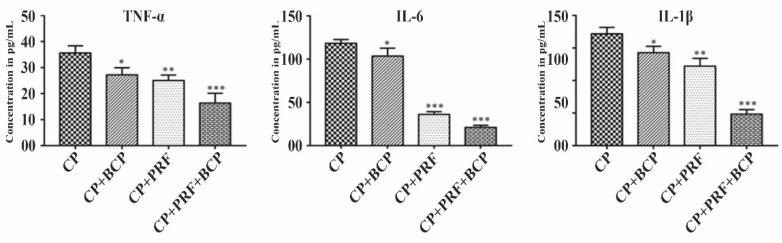
Effect of PRF/BCP on TNF-α, IL-6 and IL-1β. Results are expressed as the mean ± SD. The comparisons were made among CP, CP + BCP, CP + PRF, and CP + PRF/BCP combination. (* *p* < 0.05; ** *p*< 0.01; *** *p*< 0.001).

**Figure 3 molecules-26-06578-f003:**
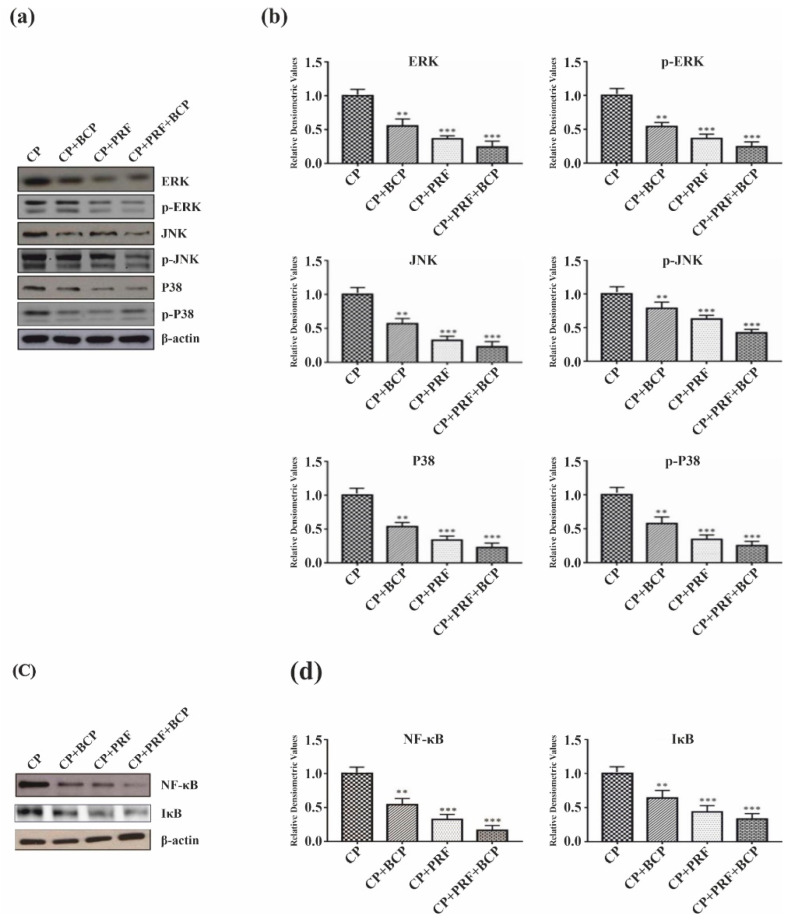
Effect of PRF/BCP on *MAPK*/*NF-kB* pathway. Western blot analysis of *MAPK* molecules (*ERK, p-ERK*, *JNK*, p-*JNK* and *P38, p-P38*) were blotted and normalized with β-actin. (**a**) Representative blot image. (**b**) Relative densitometric analysis in histograms. *NF-kB* and *IKB* molecules were blotted and normalized with β-actin. (**c**) Representative blot image. (**d**) Relative densitometric analysis in histograms. Results are presented by mean ± SD. Comparisons were made among CP and BCP, CP and PRF, and CP and PRF/BCP combination. (** *p*< 0.01; *** *p*< 0.001).

**Figure 4 molecules-26-06578-f004:**
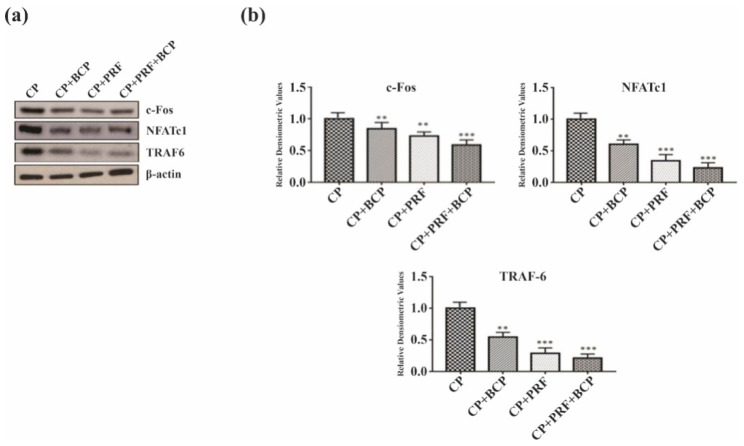
PRF/BCP inhibits osteoclastogenic transcription factors. Results are expressed as the mean ± SD. Western blot analysis of osteoclast pathway molecules (*c-Fos*, *NFATc1* and TRAF 6) was blotted and normalized with β-actin. (**a**) Representative blot image. (**b**) Relative densitometric analysis in histograms. Results are expressed as mean ± SD. Comparisons were made among CP and BCP, CP and PRF, and CP and PRF/BCP combination. (** *p* < 0.01; *** *p* < 0.001).

**Figure 5 molecules-26-06578-f005:**
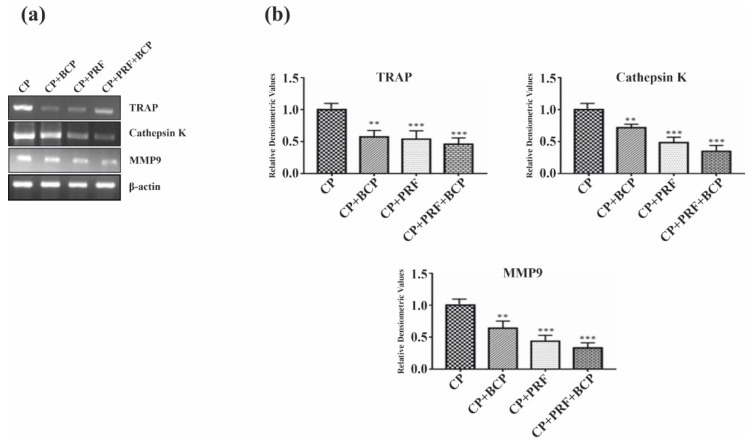
PRF/BCP inhibits osteoclast marker genes. RT-PCR analysis of osteoclastogenic genes *TRAP*, cathepsin K and MMP-9. (**a**) Representative blot image. (**b**) Relative densitometric analysis in histograms. Results are expressed as mean ± SD. Comparisons were made among CP and BCP, CP and PRF, and CP and PRF/BCP combination. (** *p* < 0.01; *** *p* < 0.001).

**Table 1 molecules-26-06578-t001:** Primer sequences used in real-time PCR.

Gene	Forward Primer Sequence (5′ to 3′)	Reverse Primer Sequence (5′ to 3′)	Size (bp)
*TRAP*	CCAATGCCAAAGAGATCGCC	TCTGTGCAGAGACGTTGCCAAG	216
*CAT-K*	CCGCAGTAATGACACCCTTT	AAGGCATTGGTCATGTAGCC	258
*MMP-9*	CTCTGGAGGTTCGACGTG	GTCCACCTGGTTCAACTCAC	183
*β-actin*	TCACCCACACTGTGCCCATCTACGA	CAGCGGAACCGCTCATTGCCAATGG	141

*TRAP*: tartrate resistant acid phosphatase; *CAT-K*: cathepsin K; *MMP-9*: matrix metalloproteinase 9; *β-actin*: beta-actin.

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
