# Peer review of "Synergistic Effect of Biphasic Calcium Phosphate and Platelet-Rich Fibrin Attenuate Markers for Inflammation and Osteoclast Differentiation by Suppressing NF-κB/MAPK Signaling Pathway in Chronic Periodontitis"

_molecules, 2021, doi:10.3390/molecules26216578_

Round 1

Reviewer 1 Report

The manuscript could almost be published in its present state, but the material and methods in the statistics section need to be completed.

The authors performed an ANOVA and then made comparisons between pairs of groups. They have not indicated the statistical method with which they have used for the comparisons.

The authors should include the statistical method used in the comparisons.   As you have performed the analysis with IBM SPSS, it should be an a posteriori contrast. (usually by Scheffe, Tukey, or Bonferroni methods)

Author Response

REVIEWER 1

The manuscript could almost be published in its present state, but the material and methods in the statistics section need to be completed.

 Question: 1

The authors performed an ANOVA and then made comparisons between pairs of groups. They have not indicated the statistical method with which they have used for the comparisons.

The authors should include the statistical method used in the comparisons.   As you have performed the analysis with IBM SPSS, it should be an a posteriori contrast. (usually by Scheffe, Tukey, or Bonferroni methods)

Answer:

Thank you for the reviewer’s comments. The differences between groups were assessed by one way ANOVA and Dunnett’s multiple comparison tests for intergroup comparison using Graphpad Prisim 6.0 software package for windows. p <0.05 was considered to be statistically significant. The above statement has been included in the statistical analysis secession.

Reviewer 2 Report

This manuscript shows that PRF and BCP inhibit markers for inflammation and osteoclast differentiation in primary cultures of adherent monocytes isolated from peripheral blood.

The manuscript is well organized and easy to read. Despite that there are several shortcomings associated with the study.

  1. All individuals were diagnosed for active PD! Will monocytes isolated from these individuals be more prone to differentiate towards OC than monocytes from periodontally healthy individuals? Control experiments with cells from healthy individuals or a cell line should have been helpful to evaluate the generalization of the findings.
  2. The adherent monocytes are not further characterized in the study. Do they contain other cells than CD14 positive cells (any lymphoid cells).
  3. Several markers involved in osteoclast differentiation have been analysed, however, presence of osteoclasts or activity of these cells have not been addressed. Throughout the whole manuscript, adjust to “markers for inflammation” and “markers for osteoclast differentiation.
  4. Several of the markers for inflammation and OC-differentiation is downregulated in presence of PRF/BC. A cytotoxicity test is needed to evaluate that this effect not is a result of cytotoxicity. A dosresponse cytotoxicity test should has been very supportive to include in the study

Minor points

Line 1-4, adjust in title “attenuate markers for inflammation and osteoclast differentiation” OC differentiation and inflammation has not been addressed in this paper

Line 60-61, add ”infection induced” in accordance to reference 1.

Line 66, delete Lipopolysaccharides. There are so many different components, way write just LPS?

Line 90, ad also OPG in this interplay

95-97, add reference to this section

Line 111, locomotion? Should it be migration?

Line 122, in which plants are the compounds present?

Line 193-204, why were not the monocytes/macrophages allowed to be adapted to the culture conditions before the differentiation was induced?

Line 442-443, change TRAP-stain to TRAP-activity

Author Response

REVIEWER 2

Question: 1

All individuals were diagnosed for active PD! Will monocytes isolated from these individuals be more prone to differentiate towards OC than monocytes from periodontally healthy individuals? Control experiments with cells from healthy individuals or a cell line should have been helpful to evaluate the generalization of the findings.

Answer:

Thank you for your suggestions. The monocytes isolated from the peripheral blood of the periodontitis patients are more prone to differentiate towards osteoclast which has also been stated by Herrera BS et al. The reference is mentioned below.

Herrera BS, Bastos AS, Coimbra LS, Teixeira SA, Rossa C Jr, Van Dyke TE, Muscara MN, Spolidorio LC. Peripheral blood mononuclear phagocytes from patients with chronic periodontitis are primed for osteoclast formation. J Periodontol. 2014 Apr;85(4): e72-81. doi: 10.1902/jop.2013.130280. Epub 2013 Sep 24. PMID: 24059638.

According to the suggestions of the reviewers, the above statement has been incorporated into the inclusion and exclusion criteria of the manuscript. The isolation of the monocytes from the peripheral blood of the healthy individuals was not included in the study since the above is already been stated in the previous literature.

The main aim of the study was to assess the synergistic effect of the regenerative material on the osteoclastic activity in periodontitis and hence the focus was made mainly on the synergistic effect of the BCP/PRF on the markers of the inflammation and osteoclastic differentiation of the monocytes from periodontitis patients. The reviewer’s suggestions are well taken and will be incorporated in our future research.

(page number 4; Line number 163-165)

Question 2: The adherent monocytes are not further characterized in the study. Do they contain other cells than CD14 positive cells (any lymphoid cells).

Answer:

In the present study the monocytes were not taken into characterization since the aim was to check the differentiation of monocytes towards the osteoclast formation and the main focus was made to analyse the effect of regenerative material on the osteoclastic activity and markers of inflammation. The reviewer’s suggestion is well taken. The authors further intend to characterize the monocytes and assess the type of the monocytes towards the osteoclastogenesis.

Question 3: Several markers involved in osteoclast differentiation have been analysed, however, presence of osteoclasts or activity of these cells have not been addressed. Throughout the whole manuscript, adjust to “markers for inflammation” and “markers for osteoclast differentiation.

Answer: Thank you for your kind comments. The osteoclastic activity was not addressed in this manuscript as the main focus was on the osteoclastic genes. Reviewer’s suggestions are taken. According to reviewer’s comments, the statements such as “markers for inflammation and osteoclastic differentiation” has been adjusted and incorporated.

(page number 1; Line number 3)

(page number 3; Line number 128)

(page number 9; Line number 355)

Question 4: Several of the markers for inflammation and OC-differentiation is downregulated in presence of PRF/BC. A cytotoxicity test is needed to evaluate that this effect not is a result of cytotoxicity. A dose response cytotoxicity test should has been very supportive to include in the study

Answer : Reviewer’s comments are well taken. The cytotoxic tests have been done in our previous study. A desired concentration of BCP in 60 microgram/ml prevented osteoclastogenesis by inducing apoptosis of osteoclast. Hence, the above concentration was taken into consideration in order to assess the markers of inflammation and osteoclast differentiation.

(page number 4; Line number 181)

Kumar, A.; Mahendra, J.; Samuel, S.; Govindraj, J.; Loganathan, T.; Vashum, Y.; Mahendra, L.; Krishnamoorthy, T. Platelet-rich fibrin/biphasic calcium phosphate impairs osteoclast differentiation and promotes apoptosis by the intrinsic mitochondrial pathway in chronic periodontitis. J. Periodontol.2019, 90, 61–71, doi:10.1002/JPER.17-0306.

Minor points

Question 5: Line 1-4, adjust in title “attenuate markers for inflammation and markers for markers for markers for markers for osteoclast differentiation” OC differentiation and inflammation has not been addressed in this paper

Answer: Thank you for your comments.  The reviewer’s suggestions are well taken. The changes were made as per the authors suggestions in the title of the article.

 (page number 1; Line number 1-4)

Question 6 : Line 60-61, add ”infection induced” in accordance to reference 1.

Answer: Thank you for your comments. The infectious part of the periodontitis has been included based on the reviewer suggestions.

(page number 2; Line number 60-61)

Question 7: Line 66, delete Lipopolysaccharides. There are so many different components, way write just LPS?

Answer: As per the reviewer suggestions modification has been made

(page number 2; Line number 66)

Question 8: Line 90, ad also OPG in this interplay

Answer: As per the reviewer suggestions modification has been made in the manuscript

(page number 2; Line number 93)

Question 9: 95-97, add reference to this section

Answer: As per the reviewer suggestions modification has been made in the manuscript and the reference has been added.

(page number 2; Line number 96-97)

Question 10: Line 111, locomotion? Should it be migration?

Answer: As per the reviewer suggestions modification has been made in the manuscript. The ‘locomotion’ has been changed with ‘migration’. (page number 3; Line number 112)

Question 11: Line 122, in which plants are the compounds present?

Answer: The components explained in line 122 has been isolated from the medicinal plants such as Reseda luteola, Morin and researched. The references are given below.

Luteolin is the principal yellow dye compound that is obtained from the plant Reseda luteola, which has been used as a source of the dye.

Gutiérrez-Venegas G, Contreras-Sánchez A. Luteolin and fisetin inhibit the effects of lipopolysaccharide obtained from Porphyromonas gingivalis in human gingival fibroblasts. Mol Biol Rep. 2013 Jan;40(1):477-85. doi: 10.1007/s11033-012-2083-0. Epub 2012 Oct 11. PMID: 23054013

Morin contained in fruits, vegetables, and beverages attenuated the intestinal inflammation of acute and chronic TNBS-induced colitis by inhibition of colonic production of leukotriene B4, NO and IL-1β.

Flavonoids are one of the secondary metabolites belongs to a polyphenolic group, and are commonly found from different parts of the plant sources like fruit, vegetables, nuts, stems, seeds, flowers, tea, wine, propolis and honey.

 Wu TW, Zeng LH, Wu J, Fung KP. Morin hydrate is a plant-derived and antioxidant-based hepatoprotector. Life Sci. 1993;53(13):PL213-8. doi: 10.1016/0024-3205(93)90266-6. PMID: 8366767.

Question 12: Line 193-204, why were not the monocytes/macrophages allowed to be adapted to the culture conditions before the differentiation was induced?

Answer: Thank you for your suggestions.  The monocytes/macrophages were not cultured as the main focus was on isolation and differentiation. Based on the reviewer suggestions in our future studies the monocytes/macrophages will be  adapted to the culture conditions and will be considered for the further research protocol.

Question 13: Line 442-443, change TRAP-stain to TRAP-activity

Answer: Thank you for your suggestions. The changes have been made as per the reviewer suggestions. (page number 11; Line number 445)

Reviewer 3 Report

Very interesting article, well written and useful for future research using stem cells. 
Given the importance, especially in this period of pandemic of "mast cells" considered at the heart of the mechanisms of inflammation, especially in the oral cavity, why not consider them in a future study?
or why not investigate how they influence the factors and cytokines you mentioned?

please read: "Conti P, Caraffa A, Gallenga CE, Ross R, Kritas SK, Frydas I, Younes A, Ronconi G. Coronavirus-19 (SARS-CoV-2) induces acute severe lung inflammation via IL-1 causing cytokine storm in COVID-19: a promising inhibitory strategy. J Biol Regul Homeost Agents. 2020 Nov-Dec;34(6):1971-1975. doi: 10.23812/20-1-E. PMID: 33016027.

"Tetè G, D'orto B, Ferrante L, Polizzi E, Cattoni F. Role of mast cells in oral inflammation. J Biol Regul Homeost Agents. 2021 Jul-Aug;35(4 Suppl. 1):65-70. doi: 10.23812/21-4supp1-6. PMID: 34425662."

b)was no age or sex range selected in the inclusion criteria? if not, why not? it is known from the literature that gender and age influence the production of cytokines and mediators of infection. 

C)why hasn't the growth of osteoblasts also been tested as being influenced by these factors? or it could be a starting point for other studies. 

please read: Tetè G, D'Orto B, Nagni M, Agostinacchio M, Polizzi E, Agliardi E. Role of induced pluripotent stem cells (IPSCS) in bone tissue regeneration in dentistry: a narrative review. J Biol Regul Homeost Agents. 2020 Nov-Dec;34(6 Suppl. 3):1-10. PMID: 33386051.

Tetè G, Capparè P, Gherlone E. New Application of Osteogenic Differentiation from HiPS Stem Cells for Evaluating the Osteogenic Potential of Nanomaterials in Dentistry. Int J Environ Res Public Health. 2020 Mar 16;17(6):1947. doi: 10.3390/ijerph17061947. PMID: 32188154; PMCID: PMC7142891.

Author Response

Reviewer 3:

Very interesting article, well written and useful for future research using stem cells. 

Question 1

Given the importance, especially in this period of pandemic of "mast cells" considered at the heart of the mechanisms of inflammation, especially in the oral cavity, why not consider them in a future study?
or why not investigate how they influence the factors and cytokines you mentioned?

please read: "Conti P, Caraffa A, Gallenga CE, Ross R, Kritas SK, Frydas I, Younes A, Ronconi G. Coronavirus-19 (SARS-CoV-2) induces acute severe lung inflammation via IL-1 causing cytokine storm in COVID-19: a promising inhibitory strategy. J Biol Regul Homeost Agents. 2020 Nov-Dec;34(6):1971-1975. doi: 10.23812/20-1-E. PMID: 33016027.

"Tetè G, D'orto B, Ferrante L, Polizzi E, Cattoni F. Role of mast cells in oral inflammation. J Biol Regul Homeost Agents. 2021 Jul-Aug;35(4 Suppl. 1):65-70. doi: 10.23812/21-4supp1-6. PMID: 34425662."s

Answer: We thank the reviewer suggestions. In this era of pandemic, mast cells are one of the most important cells of defence which play a pivotal role in inflammation. In our future study we would also consider the role of mast cells in periodontal inflammation with regard to pro inflammatory cytokines.

Question 2:

  1. b) was no age or sex range selected in the inclusion criteria? if not, why not? it is known from the literature that gender and age influence the production of cytokines and mediators of infection. 

Answer:

Reviewer suggestions are well taken. Gender and age definitely influence the production of cytokines and mediators of infection. In our study 15 chronic periodontitis patients with in the age range of 35-45 were selected. The same has been incorporated into the study design section of the manuscript.

(page number 3; Line number 149-150)

(page number 4; Line number 164-170)

Question 3:

C)why hasn't the growth of osteoblasts also been tested as being influenced by these factors? or it could be a starting point for other studies. 

Answer: Reviewer suggestions were well taken. The aim of the present study was to analyse the synergistic effect of PRF and BCP on the markers of inflammation and osteoclastic marker genes. As per the reviewer suggestions, growth of osteoblasts influenced by the growth factors will be included as the next module of our research.

Round 2

Reviewer 2 Report

The revised manuscript has been substantially improved and and suggest it can now be pulished in its present form.

This manuscript is a resubmission of an earlier submission. The following is a list of the peer review reports and author responses from that submission.

Round 1

Reviewer 1 Report

Manuscript: Synergistic effect of Biphasic calcium phosphate and Platelet rich fibrin attenuates inflammation and osteoclast differentiation by suppressing NF-κB/MAPK signaling pathway in chronic periodontitis.

The study has provided a descriptive approach to the effects of BCP, PRF (alone or in combination) at the differentiation level on PBMCs derived osteoclast from periodontitis patients. However, some points should be addressed to persuade the reviewer.

Previous publication from this group has already demonstrated the role of BCP, PRF on osteoclast formation through promotion apoptosis [1].

The authors claim that “To the best of our knowledge, this is the first study to report the inhibitory and synergetic effects of PRF and bone substitute BCP on osteoclast differentiation and function.” However, because a previous publication [1] demonstrated that “PRF/BCP displayed an inhibitory role in osteoclasts formation,” the review suggests removing or modifying this sentence in the introduction.

The results reported in figure 1 are not new only support the previous results from this group, where a decrease on TRAP-positive cells with BCP, PRF and PRF/BCP compared to CP was shown [1]. The author strongly suggests removing this result or reported as a confirmation of previous publication.

To demonstrated that reduction of inflammatory cytokines is not a consequence of the toxic effect of the treatment, viability assay of osteoclast exposed to BCP, PRF and PRF/BCP in the concentrations/proportion used in the study should be provided.

In Figure 3a, the representative blot image showed p-ERK with two bands. Why is there only band in the ERK. The same issue occurred in p-JNK and JNK. Do the used antibodies detect several isoforms? Please provide the reference number of the antibodies and ng of protein used for western blot.

In the material and methods section, the authors mentioned that all the data were expressed as mean ± standard deviation (SD). However, according to the figure's legends, the results were represented as mean ± standard deviation error. Please modify accordingly and add the errors bars also to the group CP.

The statistics analysis performed in the study was t-test, an analysis used to compare the means of two groups. Since the group number is more than two, the authors should perform a non-parametric one-way ANOVA analysis.

Additionally, the number of biological and technical replicates used in the study is not shown in the material and methods section and figure legends.

Minor comments:

Foreign words and gene names should always be in italic (e.g., et al.)

Gene Accession Number for the primers is missing

Several typos’ mistakes (e.g., missing period before references, line 116 ostoeoclastic > osteoclastic)

In figure 3 © should be changed to (c)

Subtitle 2.9 Western blot analysis should be in italic

Several abbreviations are missing or do not explain the first time used (e.g., CP figure 1 legend)

Reference

[1] Kumar A, Mahendra J, Samuel S, Govindraj J, Loganathan T, Vashum Y, Mahendra L, Krishnamoorthy T. Platelet-rich fibrin/biphasic calcium phosphate impairs osteoclast differentiation and promotes apoptosis by the intrinsic mitochondrial pathway in chronic periodontitis. J Periodontol. 2019 Jan;90(1):61-71. doi: 10.1002/JPER.17-0306. Epub 2018 Sep 11. PMID: 29958327.

Reviewer 2 Report

In the manuscript entitled: “Synergistic effect of Biphasic calcium phosphate and Platelet rich fibrin attenuates inflammation and osteoclast differentiation by suppressing NF-κB/MAPK signaling pathway in chronic periodontitis”, the authors aimed to examine the involvement of the Nuclear Factor kappa-light- chain-enhancer of activated B cells (NF-kB) and mitogen-activated protein kinase (MAPK) signaling pathway in osteoclast differentiation and their modulation by PRF/BCP in chronic periodontitis.

The authors found that the potent inhibitory effect of PRF/BCP on osteoclastogenesis was evidenced by the decreased TRAP activity and the expression of transcription factors, NFATc1, c-Fos, and the osteoclast marker genes, TRAP, MMP-9, and Cathepsin-K were found to be reduced.

The authors concluded that PRF/BCP may be a promoting synergetic combination that could be used as a strong inhibitor of inflammation-induced osteoclastogenesis in chronic periodontitis.

Major comments:

In general, the idea and innovation of this study, regards analysis of a osteoclast differentiation model by suppressing NF-κB/MAPK signaling pathway in periodontitis is interesting, because the role of these factors in dentistry are validated but further studies on this topic could be an innovative issue in this field could be open a creative matter of debate in literature by adding new information. Moreover, there are few reports in the literature that studied this interesting topic with this kind of study design.

The study was well conducted by the authors; However, there are some concerns to revise that are described below.

The introduction section resumes the existing knowledge regarding the important factor linked with inflammatory mediators and growth factors associated of mediators released during periodontitis.

However, as the importance of the topic, the reviewer strongly recommends, before a further re-evaluation of the manuscript, to update the literature through read, discuss and must cites in the references with great attention all of those recent interesting articles, that helps the authors to better introduce and discuss the role of materials in oral tissues and related release of inflammatory mediators as transglutaminase 2 and IL-1B during periodontitis: 1) Isola G, Lo Giudice A, Polizzi A, Alibrandi A, Murabito P, Indelicato F. Identification of the different salivary Interleukin-6 profiles in patients with periodontitis: A cross-sectional study. Arch Oral Biol. 2021 Feb;122:104997. doi: 10.1016/j.archoralbio.2020.104997. 2) Matarese G, Currò M, Isola G, Caccamo D, Vecchio M, Giunta ML, Ramaglia L, Cordasco G, Williams RC, Ientile R. Transglutaminase 2 up-regulation is associated with RANKL/OPG pathway in cultured HPDL cells and THP-1-differentiated macrophages. Amino Acids. 2015 Nov;47(11):2447-55. doi: 10.1007/s00726-015-2039-5.

The authors should be better specified, at the end of the introduction section, the rational of the study and the aim of the study. In the material and methods section, should clarify the PRF preparation, as well as the update of the new term “periodontitis” instead of chronic periodontitis. Specify also grades and stages of periodontitis. Moreover, please more specify the scientists involved in the different stages of the study.

The discussion section appears well organized with the relevant paper that support the conclusions, even if the authors should better discuss the relationship between materials implantation and related host inflammation. The conclusion should reinforce in light of the discussions.

In conclusion, I am sure that the authors are fine clinicians who achieve very nice results with their adopted protocol. However, this study, in my view does not in its current form satisfy a very high scientific requirement for publication in this journal and requests a revision before a futher re-evaluation of the manuscript.

Minor Comments:

Abstract:

  • Better formulate the abstract section by better describing the aim of the study

Introduction:

  • Please refer to major comments

Discussion

  • Please add a specific sentence that clarifies the results obtained in the first part of the discussion

Page 10 last paragraph: Please reorganize this paragraph that is not clear

Reviewer 3 Report

Major comments:

Some data are missing :

  • Fig 1 : TRAP staining have to be shown
  • Osteoclasts in primary cell culture are very heterogeneous. The different results should be completed by using immunocytofluorescence (NFkB - IkB, MAPK pathways etc...). In the discussion, authors write " In the present study, IL-1β, IL-6, and  360  
    TNF-α were increased in the CP group suggesting an inflammatory response during the progression  of  the  disease  (Fig.  2).  " This is on overinterpretation. The overrelease of these mediators within inflammed periodontium is due to many cells such as inflammatory cells.
  • The effects of PRF + BCP on the resorption activity of osteoclasts should be tested and measured.
  • The apoptosis of the cells have to be tested and quantified.
  • The effects of PRF + BCP on osteoclasts activity have to be tested and measured.

Moreover :

Figure 2  : the induction of OC differentiation by cytokines (I-1, TNF...) is well known but are osteoclasts potent producer of these cytokines? We need references 

"Study Limitations." The authors state that "Second, there was a lack of healthy groups that could provide more information...". This means that OC from PBMCs of periodontitis patients and healthy patients is not the same: is there any data on this? 

Minor comments:

  • Ref 2 has to be modified
  • In the introduction : the hypothesis and the aim of the study are not in accordance
  • The paragraphs in the introduction can be better organized and the introduction is quite long
  • page 2 line 86 : "MAPK implicates " = MAPK are implicated in ...?
  • page 2 lines 92-93 : a reference is missing
  • Table 1 : add the amplicon lengths